# The Integration of Complex Systems Science and Community-Based Research: A Scoping Review

**Travis R. Moore** [1,*], **Nicholas Cardamone** [2], **Helena VonVille** [3] and **Robert W. S. Coulter** [4]

1 Division of Nutrition Interventions, Communication, and Behavior Change, Friedman School of Nutrition Science and Policy, Tufts University, Boston, MA 02111, USA
2 Penn Center for Mental Health, Perelman School of Medicine, University of Pennsylvania, Philadelphia, PA 19104, USA; nicholas.cardamone@pennmedicine.upenn.edu
3 Health Sciences Library System, University of Pittsburg, Pittsburgh, PA 15260, USA; helenavonville@pitt.edu
4 Department of Behavioral and Community Health Sciences, School of Public Health, University of Pittsburgh, Pittsburgh, PA 15260, USA; robert.ws.coulter@pitt.edu
* Correspondence: travis.moore@tufts.edu

**Abstract:** Complex systems science (CSS) and community-based research (CBR) have emerged over the past 50 years as complementary disciplines. However, there is a gap in understanding what has driven the recent proliferation of integrating these two disciplines to study complex and relevant issues. In this review, we report on the results of a scoping review of articles that utilized both disciplines. After two levels of reviewing articles using DistillerSR, a web-based platform designed to streamline and facilitate the process of conducting systematic reviews, we used two forms of natural language processing to extract data. We developed a novel named entity recognition model to extract descriptive information from the corpus of articles. We also conducted dynamic topic modeling to deductively examine in tandem the development of CSS and CBR and to inductively discover the specific topics that may be driving their use in research and practice. We find that among the CSS and CBR papers, CBR topic frequency has grown at a faster pace than CSS, with CBR using CSS concepts and techniques more often. Four topics that may be driving this trend are collaboration within social systems, business management, food and land use and knowledge, and water shed management. We conclude by discussing the implications of this work for researchers and practitioners who are interested in studying and solving complex social, economic, and health-related issues.

**Keywords:** complex systems science; community-based research; scoping review; natural language processing; complexity science; community-engaged research; community-based system dynamics



## 1. Introduction

The integration of complex systems science (CSS) and community-based research (CBR) has gained traction in addressing persistent and intricate challenges [1,2]. CSS is a field dedicated to understanding the dynamic interactions among diverse elements (and actors) and their environments, resulting in complex patterns of phenomena over time. It encompasses a wide range of concepts and analytical techniques that aim to explore the intricate interplay of actors at various scales, each driven by distinct motivations and priorities. Unlike simplistic approaches, CSS acknowledges the multi-faceted nature of actor systems, characterized by numerous interconnected components and pathways that collectively shape the system's behavior. These systems exhibit rich variation and outcomes that cannot be attributed to a single mechanism alone. To capture this complexity, CSS identifies several general properties, such as heterogeneity, emergence, and interdependence [3,4]. Recognizing the value of these properties in unraveling complex phenomena, numerous fields, including CBR, have adopted this framework to tackle challenging and multifaceted issues.

CBR is a collaborative and participatory approach to inquiry that involves partnerships between researchers and community members in order to address shared concerns and produce socially relevant knowledge for the purposes of achieving equity [5,6]. Unlike a rigid methodology, CBR, which we operationalized to include community-based participatory research, is organized around a set of guiding principles [7]. Firstly, it acknowledges the community as a primary unit of identity. Secondly, it leverages community strengths and resources. Thirdly, it encourages cooperative and fair partnerships across all research stages, advocating for shared power. Fourthly, it underscores mutual learning and skill development among all collaborators. Finally, it advocates for research as a sustained effort dedicated to a lasting impact, especially in confronting concerns related to race, ethnicity, discrimination, and social stratification.

These principles provide researchers with a framework to conduct research processes that embrace a community-engaged approach, where residents have equal power in shaping the research agenda and resource allocation [7]. It is crucial to recognize the diverse realities within the concept of "community". Hence, we adopt MacQueen and colleagues' (2001) identification of five core elements: locus (a sense of place), sharing (common interests and perspectives), joint action (coherence and identity), social ties, and diversity [8]. In our review, we adopt a broader understanding of CBR principles, including the related (and often conflated) approach called community-based participatory research [9], to emphasize the inclusion of community members and stakeholders in any aspect of the research process.

In recent years, there has been a growing trend among researchers to integrate CSS and CBR to address seemingly intractable concerns, such as childhood obesity, tobacco use, and HIV [10–12]. For instance, in the Catalyzing Communities project at Tufts University, researchers collaboratively merged group model building, social network analysis, and computational modeling to evaluate the diffusion of knowledge and engagement in childhood obesity prevention from a small community coalition to the broader community, with the goal of creating and sustaining local changes in policies and practices that influence child nutrition and physical activity [13–15]. This innovative approach integrated CSS concepts and analytical techniques with CBR principles and methods, and has recently been cited as an exemplar systems-informed implementation of a science project [16].

While this example pertains to nutrition and public health, numerous instances can be found in diverse fields such as environmental science [17,18], healthcare [19], medicine [20], and food systems [21], among others. More recently, journal Special Issues have been dedicated to the overlap between CSS and CBR [22]. Despite the growing utilization of these combined approaches, their deployment and impact remain relatively unexplored, with no existing scoping review encompassing both CSS and CBR [4]. Exploring the overlap between CSS and CBR can uncover synergies and complementarities between these two fields, leading to more integrated and holistic solutions to complex problems. Exploring the overlap between these fields can also inspire the development of novel research methodologies and tools that are tailored to address complex real-world challenges. A comprehensive search conducted on 1 September 2019 across various databases yielded no scoping protocol or review specifically focusing on the integration of CSS and CBR, although a related systematic review explored community participation in health systems research [23]. However, this review did not treat CSS and CBR as distinct bodies of literature and primarily concentrated on health systems.

In this scoping review, our central questions are as follows (1) To what extent do CSS research and CBR overlap? (2) Which topics within these disciplines seem to be driving this overlap in research? Naturally, a scoping review of this kind (e.g., not bounded by time or specific sub-discipline) covers a large body of work. Thus, we developed novel scoping review methods to further explore the CSS and CBR corpus of articles. By conducting a scoping review using the traditional methods of multi-level screening and data extraction combined with a novel implementation of natural language processing on the screened corpus, we hope to provide insights into each research question. We present our research

findings addressing these two primary questions and derive conclusions regarding their significance for researchers and practitioners seeking to expand their knowledge and apply it to their own work in related areas.

While publishing a scoping review protocol is not mandatory, it is increasingly considered best practice to make it publicly available before starting a scoping review. Unlike our previous publication outlining the protocols used for this scoping review [4], this manuscript delves into the substantive results of the scoping review. By presenting our findings, we contribute valuable insights into the key concepts within CSS and CBR. In this manuscript, in addition to reporting our findings, we also document the development of novel scoping review methods to analyze the corpus of articles included in this review.

## 2. Materials and Methods

As reported in our published scoping review protocol [4], this scoping review followed the reporting guidelines as set forth by the Preferred Reporting Items for Systematic Reviews and Meta-Analyses for Scoping Reviews (PRISMA; [24]) and JBI's Reviewer's Manual [25], and was assessed for quality using the AMSTAR 2.0 checklist [26].

### 2.1. Inclusion Criteria

Table 1 provides an overview of the inclusion criteria employed in our study. The eligibility criteria and analytical methods were established beforehand. The objective of our scoping review was to gain insights into the distinct yet interconnected domains of CSS and CBR. To fulfill this objective, we formulated two key review questions: "How are complex system science concepts and/or strategies incorporated in the article?" and "Does the study involve stakeholders at any stage"? Hence, the inclusion criteria for this scoping review defined the specific study attributes that helped us to achieve our primary goal of understanding how CSS and CBR have co-evolved.

**Table 1.** Inclusion criteria.

| Category | Criteria |
| --- | --- |
| Types of participants | Human subjects with a focus on those who have a stake in the research process (e.g., stakeholders, changemakers, interest holder, etc.) |
| Fields | Any field |
| Concepts | Complex systems science concepts and/or techniquesCommunity-based research concepts and/or approaches |
| Outcomes | Any |
| Language | English |
| Context | Any |
| Types of Studies | Scholarly sources |

### 2.2. Types of Participants

Our scoping review aimed to explore the intersection and practical alignment of CSS and CBR. Consequently, there were no predefined criteria for selecting participants in this review, except for the inclusion of human subjects. Given the broad spectrum of studies considered, our primary focus centered on articles in which researchers or practitioners engaged stakeholders. In this context, stakeholders encompass individuals with a vested interest or concern related to the research topic or its implications, adopting a more contemporary view that includes terms like changemaker or interest holder, as some practitioners prefer these over "stakeholder" due to its association with a capitalist mindset. It is essential to note that "stakeholder" extends beyond scientific manuscript authors, necessitating the involvement of external community members in any phase of the research process for inclusion in this review.

### 2.3. Concept

Our scoping review focused on key concepts within the disciplines of CSS and CBR. CSS encompasses a diverse array of concepts and analytical techniques, aiming to explore the structures and behaviors of actors across various scales [27,28]. These concepts, detailed in Table 2, complement our understanding of complex systems techniques, which range from computational methods like natural language processing [29] and machine learning to simulation techniques such as agent-based modeling [30]. Additionally, practical and community-driven approaches, such as causal-loop diagramming [28] and group model building [31], contribute to measuring system dynamics. See Moore and colleagues' (2021) scoping review protocol for a full list of concepts as search terms used to define an initial search of the literature [4].

**Table 2.** Complex systems properties and explanations.

| Complex System Properties | Property Explanation |
| --- | --- |
| Individuality * | CSs are often multi-level and driven by the decentralized, local interaction of constituent parts. Each level is composed of autonomous actors who adapt their behavior individually. |
| Heterogeneity * | Substantial diversity (goals, rules, constraints, etc.) among actors at each level. |
| Interdependence * | CSs usually contain many interdependent interacting pieces, connected across different levels with feedback and nonlinear dynamics. |
| Emergence * | CSs are often characterized by emergent, unexpected phenomena—patterns of collective behavior that form in the system are difficult to predict from separate understanding of each individual element. |
| Tipping * | CSs are also often characterized by tipping or the impacts caused by small changes that can seem out of proportion. |
| Nonlinearity ** | Sensitivity to initial conditions; small actions can have large consequences (see tipping). |
| Dynamical ** | Interaction within, between, and among systems and subsystems are rapidly changing. |
| Adaptive ** | Interacting elements and agents respond and adapt to each other so that what emerges and evolves is a function of ongoing adaptation among both interaction elements and the responsive relationships interacting agents have with their environment. |
| Uncertain ** | Process and outcomes are unpredictable, sometimes uncontrollable, and many times unknowable in advance. |

Note. Adapted from Moore (2021); * denotes concepts from Hammond (2009); ** denotes concepts from Patton (2010).

CBR serves as a bridge between science and practice through community engagement, stakeholder involvement, and social action, all with the overarching goal of fostering social justice and equity [6,32]. Inclusivity is a central tenet of CBR, emphasizing the involvement of stakeholders throughout the research process, from design to dissemination. In our review, we examined the depth and duration of stakeholder engagement, along with the methods employed for such engagement. While stakeholder empowerment was not an inclusion criterion, we explored its significance, defining it as the extent to which stakeholders are involved in and control the research process [33]. Furthermore, we considered capacity building as an outcome of CBR projects, referring to the process through which individuals and organizations enhance their skills, knowledge, and resources to effectively carry out their responsibilities [34].

### 2.4. Context

We examined studies globally, published in either English or Spanish, without restricting the socio-cultural context. Given the nature of CBR, our investigation encompassed

diverse community settings. Our scoping review deliberately avoided limiting the search based on any specific contextual constraints.

### 2.5. Types of Studies

Our reviewers assessed scholarly sources, including published research such as primary research studies, systematic reviews, meta-analyses, letters, guidelines, and websites. Notably, given our inclusion criterion centered around stakeholder involvement, primary research studies constituted a substantial proportion of the studies under review. Although our initial scoping review encompassed systematic reviews, systematic reviews lacking stakeholder participation in project design or execution did not meet our inclusion criteria and were consequently excluded from our review.

### 2.6. Search Strategy

Medline (Ovid) was initially searched using a combination of MeSH terms and terms found in the title, abstract, and keyword fields by a health sciences librarian with systematic review training (see Figure 1). Syntax and terminology were then adapted for Embase (Elsevier), PsycINFO (Ovid), AGRICOLA (Ovid), ERIC (EBSCO), Academic Search Premier (EBSCO), and Web of Science (Clarivate). The initial searches were conducted in July and August 2019. An updated and revised search of all databases was completed in January and February 2020. Strategies and search dates for each database are available in Moore and colleagues' 2021 scoping review protocol [4]. EndNote (Clarivate) was used initially to store all citations found in the search process and to check for duplicates. They were then uploaded into DistillerSR (Evidence Partners, Ottawa, ON, Canada). Search strategies and results were tracked using an Excel workbook designed specifically for this purpose [4]. Data related to the search as well as the search strategies are available from the corresponding author upon request.

### 2.7. Study Selection

The screening and comprehensive review of texts were conducted using DistillerSR. Before commencing the review of titles and abstracts, all participants underwent training on DistillerSR's usage and familiarized themselves with the project's objectives, eligibility criteria, and exclusion criteria. The authors and two student scholars independently screened the titles and abstracts of potential articles for inclusion, ensuring a blinded approach to journal titles. The interrater reliability score, represented by the kappa statistic, was calculated at 6.7, indicating substantial agreement among coders [35]. Any disagreements were addressed by the first author, who regularly sought and provided feedback on both unique and common disagreements.

A similar two-rater screening process was employed for reviewing full-text articles. Before initiating the review of full-text articles, each study underwent scrutiny on Retraction Watch (http://www.retractionwatch.com, accessed on 15 January 2020). Furthermore, an additional search was conducted for each study in PubMed using a retraction/correction database search filter (http://bit.ly/pubmed-filters, accessed on 1 February 2020) to ensure the study's inclusion and the accuracy of the data used for analysis. For transparency, a list of excluded citations at each stage is available upon request from the first author.

### 2.8. Data Extraction and Analysis

To examine how CSS and CBR disicplines have overlapped with one another, we employed natural language processing techniques, specifically dynamic topic modeling and named entity recognition. Dynamic topic modeling can be viewed as a magnifying glass for understanding the content of a collection of documents [36,37]. It can help uncover the underlying topics or themes that emerge from the text by analyzing patterns in the words and their relationships within the documents. In many cases, it is a way to identify the main ideas or subjects that the authors are discussing in their papers. This approach enabled us to explore both the predefined topics derived from existing theories and the

novel topics that emerged from the data themselves. On the other hand, named entity recognition is a technology that helps computers to understand and identify specific things, such as names of people, places (e.g., country of study sites), dates (e.g., publication year), organizations, and more in text [38,39]. Think of named entity recognition as a tool that reads through a story or article and picks out important information. For example, if you are reading a story, named entity recognition can spot and highlight the names of characters, locations where the story takes place, specific dates or numbers, and even names of companies or organizations.

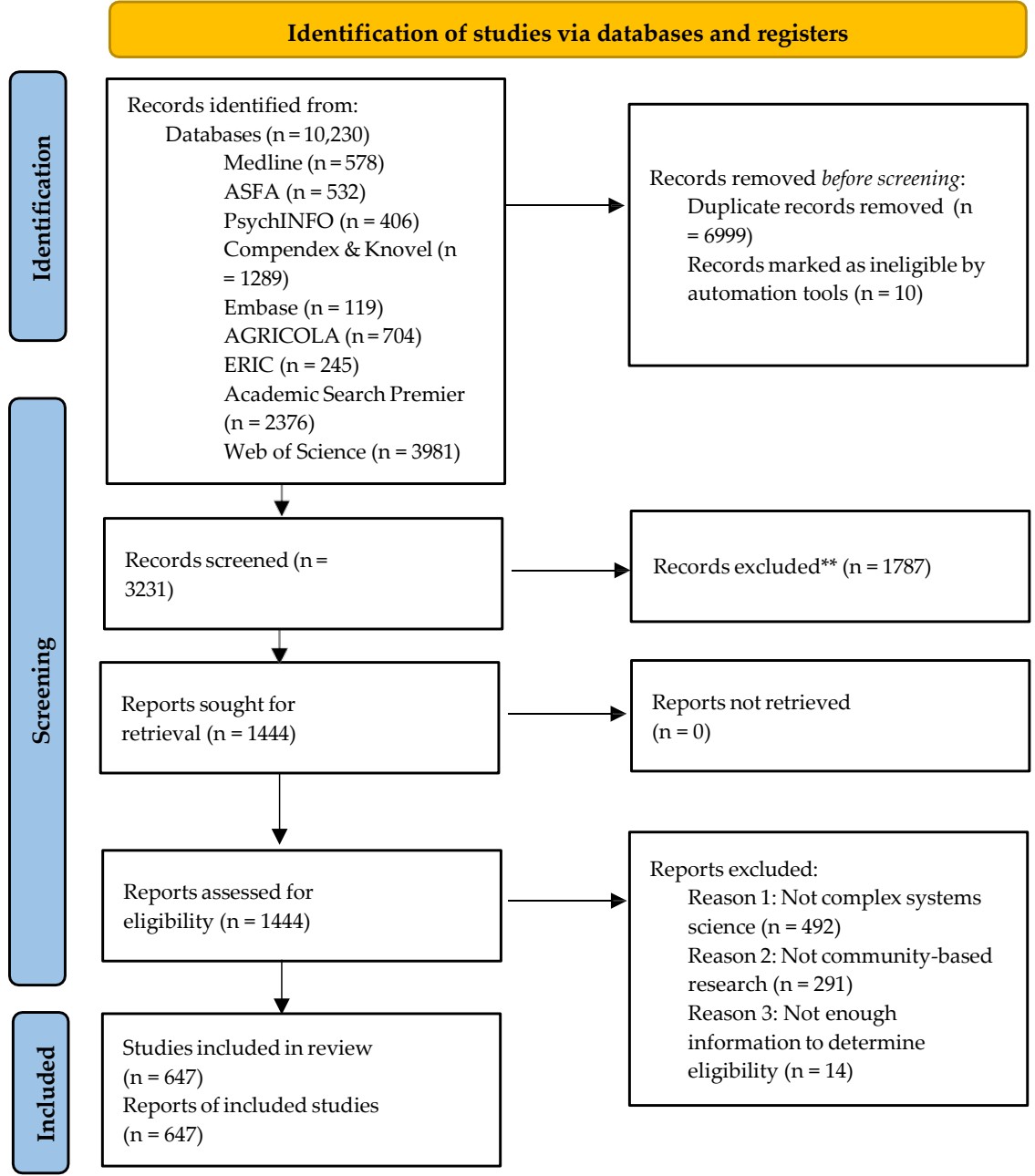

**Figure 1.** PRISMA flow diagram for scoping reviews.

2.8.1. Named Entity Recognition

In addition to dynamic topic modeling [40], we employed named entity recognition [41], an innovative large-scale information extraction technique, to identify the most frequently mentioned entities such as countries, academic disciplines, and CSS techniques

within each article. While there have been recent innovations in large-scale information extraction through techniques such as named entity recognition and relation extraction in the biomedical literature [42,43], this is the first documented implementation of named entity recognition for large-scale information extraction in the CSS and CBR literature.

Before implementing the named entity recognition model, we undertook preparatory steps to clean the text data, eliminating author names, reference pages, stop words, paragraphs with pre-specified words (i.e., "abstract", "bibliography", "mesh", "objectives", "ethics", "declarations", "acknowledgements", "references", "works cited"), and non-essential information like DOI details, web links, reference numbers, and page numbers. We also removed sentences with terms that could have created noise within the data, such as "https:", "conference of", "conference:", "conference.", "society", "(ed.)", "sage publications", "reference number:", "PhD", "PO Box", and "Elsevier", among others. We removed sentences with any regex patterns that would appear in a reference (e.g., a digit followed by a dash followed by a digit or a digit followed by a parenthetical around a digit). The character count of sentences had to be over 60. After cleaning the data, the top three entities for each article in each entity category were ranked by the total count and penalized by the distance from the median page; if there was more than one country, discipline, or technique pulled from the article, we dropped any instance where these entities were mentioned only once to remove spurious cases.

Leveraging the "spacy" module in Python, a custom named entity recognition model was created. We produced two metrics from the model: (1) a raw count of the top five most mentioned unique entity terms for each article in each entity category and (2) a count of the top three terms for each article weighted by its closeness to the median page number of the article.

### 2.8.2. Dynamic Topic Modeling

Dynamic topic modeling was utilized to systematically investigate themes and concepts present in the corpus of research manuscripts [40]. This method allowed us to deduce and induce the underlying topics and themes within and across the documents. By analyzing word patterns and relationships, dynamic topic modeling revealed both predefined and emergent topics from the dataset.

The process began with the exporting of screened manuscripts from DistillerSR, aggregating them into a consolidated file for analysis in the R programming language. Leveraging the "tidytext" package in R, we preprocessed the text corpus, filtering out irrelevant words. Additionally, several iterations of filtering were performed to refine the corpus, excluding commonly occurring but insignificant words like "et" and "al." Following the preprocessing steps, the corpus was converted into a document-term matrix, allowing for the extraction of parameters using multiple sensitivity models to determine the number of topics suitable for the machine learning algorithm. This approach is particularly useful when the number of topics is not theoretically motivated a priori. We used the "FindTopicsNumber" function using the built-in metrics from CaoJuan (2009) [44], Arun (2010) [45], and Deveaud (2014) [46] to determine the number of topics. From these analyses, we performed dynamic topic modeling using the "TopicModels" package in R, setting *k* to 10 for the number of topics that we wanted to model. Finally, a temporal resolution analysis was performed to determine the granularity of the time scale for topic distributions [47]. The temporal resolution of one year showed the best stability for topics.

The mathematical formulation of dynamic topic modeling involves defining a joint distribution over words, topics, and time slices. The joint probability of a document collection, topic assignments, topic distributions, and time stamps can be expressed as:

$$[p(w,z,\theta,\beta,t) = \prod_{d=1}^{D} p(\theta_d) \prod_{n=1}^{N_d} p(z_{dn}|\theta_d) p(w_{dn}|z_{dn},\beta_{t_{dn}}) p(t_{dn})],$$

representing the joint probability distribution of the entire corpus, where $w$ represents all words, $z$ represents all topic assignments, $\theta$ represents document-specific topic distributions, $\beta$ represents topic-specific word distributions, and $t$ represents time stamps. The product over $D$ documents and $N_d$ words in each document denotes the probability of observing the topic assignments, word occurrences, and time stamps given the respective distributions. Specifically, $p(\theta_d)$ represents the probability of the topic distribution in document $d$, $p(z_{dn}|\theta_d)$ represents the probability of the topic assignment for word $n$ in document $d$ given the document-specific topic distribution, $p(w_{dn}|z_{dn},\beta_{tdn})$ represents the probability of observing word $n$ in document $d$ given its assigned topic and the word distribution at the corresponding time slice (e.g., within each year), and $p(t_{dn})$ represents the probability of the time stamp of word $n$ in document $d$. This formula encapsulates the joint distribution of various components in dynamic topic modeling, incorporating both topic evolution over time and word generation within documents.

## 3. Results

The initial search yielded a total of 3240 articles, reports, and publications. After applying inclusion and exclusion criteria by reading abstracts as a level one review, 1453 studies were included in the second-level review. The second-level review, which included a full read of each manuscript, produced 647 manuscripts that included both CSS and CBR.

### 3.1. Describing the Corpus

Without distinguishing between CSS and CBR, named entity recognition was able to parsimoniously identify countries, academic disciplines, and CSS techniques within the corpus of articles. As seen in Figure 2, the number of countries deploying CSS and/or CBR in their research has increased. From 1990 to 1999, Nepal, Netherlands, Australia, Japan, and Liberia were the most frequently mentioned countries. The model detected eight distinct countries from the articles in this period. For example, Australian researchers conducted conceptual modeling workshops with Nepalese farmers to develop the livestock sector [48]. Venix (1996, 1999) employed group-model building with a Dutch government agency [49,50]. From 2000 to 2009, New Zealand, Thailand, Canada, Australia, and Mexico were the most frequently mentioned countries. Over 75 distinct countries on six continents were mentioned in articles in this period. From 2010 to 2019, Australia, Canada, New Zealand, Thailand, and Ghana were the top five countries mentioned. Over 145 distinct countries on six continents were mentioned in articles in this period.

The named entity recognition model returned 2656 entities related to academic disciplines from 638 articles. As seen in Figure 3, the term "management" was among the top five discipline entities in 38.2% of the articles, followed by "government" (19.6%), "education" (18.5%), "agriculture" (16.2%), and "design" (16.2%). Among terms associated with complex systems science, "simulation" appeared in the top five terms of 10.9% of the articles, followed by "system dynamics" (6.5%), "action research" (4.4%), "complex systems" (3.4%), "social network analysis" (3.3%), and "systems science" (0.9%).

Finally, the named entity recognition model returned 330 entities related to complex systems science techniques in 249 articles. As seen in Figure 4, the following complex systems named entity techniques appeared most frequently: "system dynamics" (35.3%), "systems thinking" (25.7%), "social network analysis" (20.5%), "network analysis" (16.5%), "causal loop diagram" (10.4%), "participatory model" (3.6%), "Markov (chain)" (2.4%), "multi-agent systems" (2.4%), "agent-based model" (2.4%), and "fuzzy cognitive map" (2.4%).

## Study Location Map

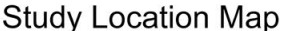

**Figure 2.** Cumulative country mention frequency 1990–2019. Note. Cumulative sum of the raw count of mention frequency extracted from articles published in 1990–1999, 2000–2009, and 2010–2019.

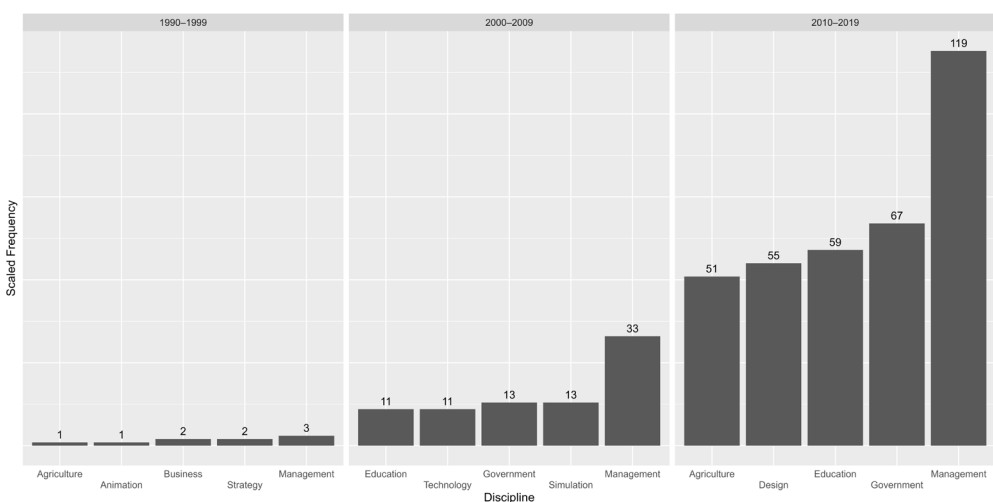

**Figure 3.** Frequency of academic discipline terms 1990–2019. Note. Most frequent terms from 1990 to 1999, 2000 to 2009, and 2010 to 2019 based on the terms extracted from each article. Article terms were ranked by their frequency and closeness to their respective article's median page (i.e., approximately the methods section) and the top three (at most) were extracted from each article.

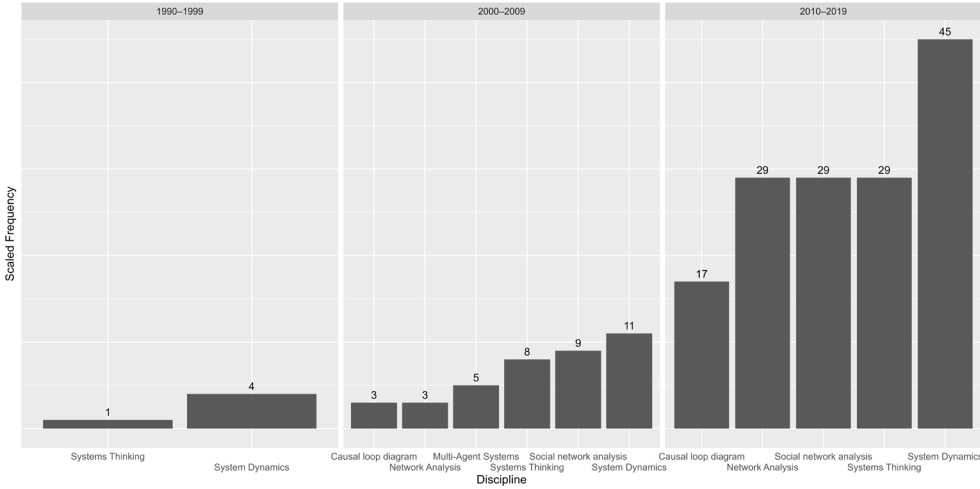

**Figure 4.** Frequency of complex systems science techniques terms 1990–2019. Note. Most frequent terms from 1990 to 1999, 2000 to 2009, and 2010 to 2019 based on the terms extracted from each article. Article terms were ranked by their frequency and closeness to their respective article's median page (i.e., approximately the methods section) and the top three (at most) were extracted from each article.

### 3.2. Tracing Complex Systems Science and Community-Based Reseach over the Last Century

The initial results of the dynamic topic modeling analysis revealed that, as seen in Figure 5, community-based research topics had a higher initial count compared to complex systems science topics, suggesting that it was a more prevalent theme among the CSS-CBR corpus. This finding illustrates that CBR topics are discussed more frequently than CSS topics within articles.

Furthermore, as the analysis progressed, it became evident that community-based research topics demonstrated a higher rate of growth compared to complex systems science. This observation suggests that community-based research not only had a higher initial presence but also experienced a more substantial increase over time within the corpus.

Figure 6 displays a heatmap illustrating the increasing overlap between CBR and CSS topics over time. The heatmap highlights the most substantial increases in overlap during specific time periods, namely from 2014 to 2015 and from 2018 to 2019. These intervals

stand out as periods when the two disciplines experienced a substantial convergence in terms of shared topics.

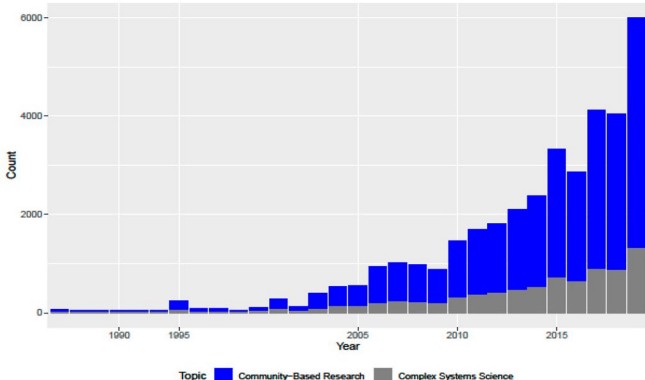

**Figure 5.** Count of distinct community-based research and complex systems science topics distributed over time. Note. "Count" is the number of times the dynamic topic model identified a topic within CBR or within CSS, within and across all included articles. For example, each article could have more than one CBR topics identified.

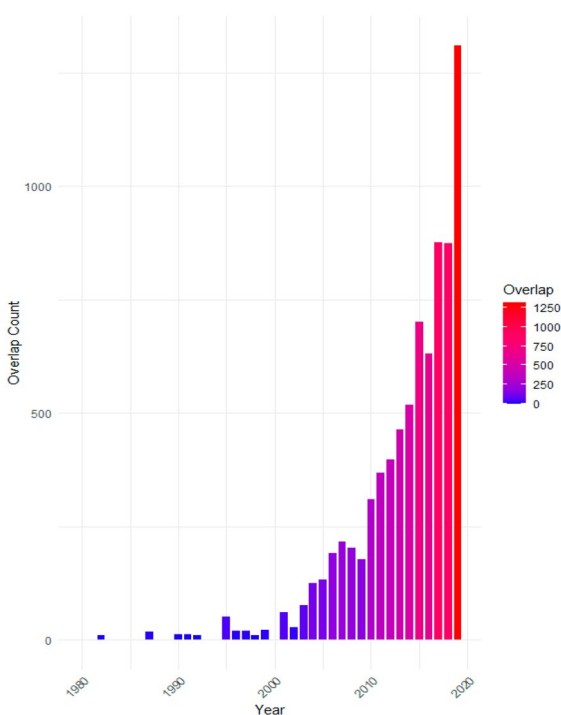

**Figure 6.** Heatmap of topical overlap over time. Note. "Overlap Count" is the number of times the dynamic topic model identified an overlap between CSS-CBR topics, within and across all included articles. The total count of topic overlap can therefore exceed the total number of articles included in the corpus.

Upon closer examination, the figure reveals that there was little to no overlap between community-based research and complex systems science topics until the late 1990s and early 2000s. This observation suggests that the integration between the two disciplines started to gain traction and become more prominent around that time.

However, the figure also indicates a decrease in topical overlap between the years 2007 and 2009, as well as between 2015 and 2016. These periods exhibit a slight divergence or reduced alignment between the two fields in terms of shared topics and concepts.

Overall, Figures 5 and 6 demonstrate a general trend of increasing overlap between community-based research and complex systems science over the examined time frame.

The heatmap visually represents the growing intersection and mutual incorporation of concepts and themes from both disciplines. This increasing overlap signifies the ongoing integration of complex systems thinking and methodologies for community-based research, highlighting the potential for interdisciplinary collaborations and the advancement of understanding and addressing complex social issues.

### 3.3. Inductive Topics Driving CBR and CSS Overlap

The dynamic topic modeling analysis of the manuscript corpus revealed 10 distinct topics that emerged from the data. Table 3 includes the 10 identified topics, along with a brief description of each topic's focus. This table serves as a reference for understanding the different thematic areas covered by the corpus of manuscripts. These topics encompassed a range of research areas, including studies on system properties, features, and processes; the modeling of systems in business and management contexts; social systems involving stakeholders and collaborative groups; modeling related to health behavior and policy; and research on social change processes, among others.

**Table 3.** Top ten topics over time.

| Topic Number | Topic Theme | Topic Words |
| --- | --- | --- |
| 1 | Research related to system properties, features, and processes | System, process, research, feature, properties, systems, model, researcher, feature |
| 2 | Research and modeling related to community organizing, mobilizing, and issues related to power | Systems, modeling, mobilize, capital, systems, organize, community, research |
| 3 | Research related to social systems involving stakeholders and other groups where collaboration takes place | Partnerships, individual, research, system, stakeholders, research, group, collaboration |
| 4 | Modeling of social systems related to community health where projects usually use participatory methods | Model, social, stakeholders, systems, health, social, systems, approach, community, participatory |
| 5 | Research and modeling related to social, developmental, and behavioral processes and approaches within the health field | Social, developmental, process, behavior, research, approach, health, model, dynamic |
| 6 | Research and modeling related to water management systems in context of community, networks | Water, management, group, community, network, system, model, one, modeling, shed |
| 7 | Processes related to food and land use and knowledge | Process, study, use, based, food, nutrition, knowledge, useful, land, also |
| 8 | Models, processes, and analyses related to business management systems that involve networks and local communities | Business, network, community, participants, model, management, local, process, system, analysis |
| 9 | The use of data to inform, influence, or study health policy | Use, policy, data, health, policies, community, different, study, well |
| 10 | Research related to social change processes and analysis | Change, research, social, analysis, process, changes, organizing, change, systems, case, move |

Additionally, Figure 7 visualizes the relative proportion of these topics from 1975 to 2019. This figure demonstrates how the 10 topics varied in their appearance within the corpus over time. Overall, this figure illustrates an increasing trend in the occurrence of these topics, with a few exceptions. This indicates that the topics became more prevalent in the literature as time progressed, suggesting a growing interest and engagement with these areas of research.

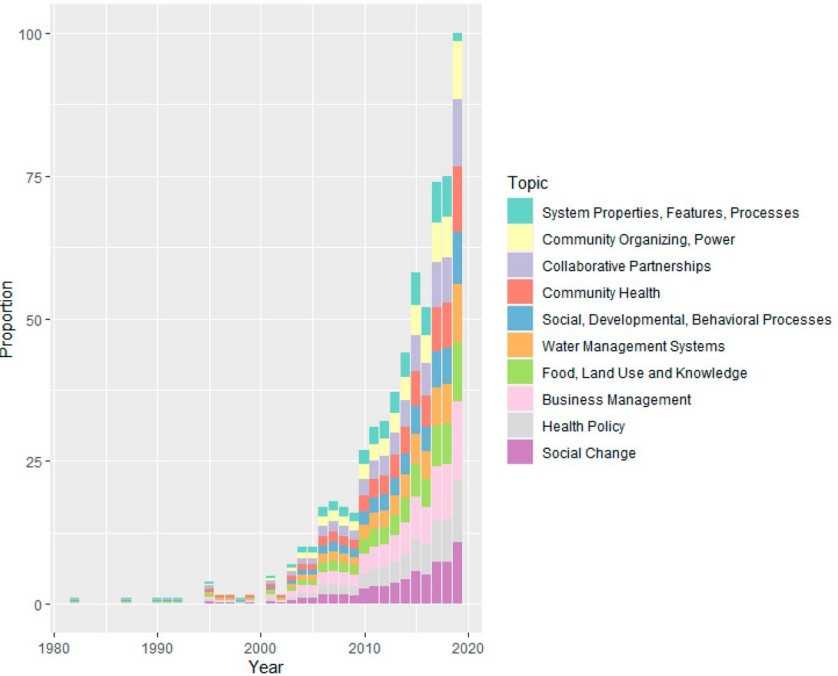

**Figure 7.** Proportion of top 10 topics each year distributed over time.

This figure also highlights four topics that seem to be driving the integration of CSS and CBR. These topics include business management systems, processes involving food and land use and knowledge, collaboration within social systems, and water shed management systems. This figure suggests that the decrease in the popularity of the topic of social change processes could potentially be a contributing factor to the decrease in overlap observed during the period from 2007 to 2009 shown in Figure 7.

Moreover, this figure reveals that in the early 1990s, the integration of CSS and CBR was primarily driven by topics related to social systems and social change processes. It was not until the late 1990s and early 2000s that other disciplines started to incorporate complex systems science and community-based research into their work. This finding indicates that while the recent increase in overlap is driven by economic and public health-related disciplines, the initial convergence between community-based research and complex systems science originated from the social sciences.

## 4. Discussion

This scoping review describes the extent to which CSS research and CBR are integrated and the topics covered within these fields, including those which seem to be driving the integration of CSS and CBR. Our results indicate substantial CSS and CBR topic overlap over time, with business management, food and lang use and knowledge, water shed management, and social systems topics substantially influencing this trend. Our discussion of the results focuses on several highlights of our results from the scoping review, extending these insights to implications for CSS, CBR, and other fields, such as public health.

Overall, the growing trend in topical overlap between CSS and CBR is encouraging, representing a shift over the last few decades toward more integrated and holistic approaches to addressing complex societal issues. By identifying prevalent research topics between CSS and CBR, such as health behavior and policy, this scoping review offers insights into the recent focus areas of CBR and CSS. As new CSS techniques emerge, the identified themes can guide and inspire future investigations that are ripe with opportunity, encouraging the development of more robust and diverse interdisciplinary approaches.

System dynamics modeling emerged as the most prevalent and rapidly growing technique within the CSS and CBR corpus, owing to its versatility in addressing a wide array of problems. However, the relatively lower prominence of other techniques, such

as agent-based models or social network analysis, does not necessarily signify that these techniques are inferior. Instead, it indicates that they are underutilized or underrecognized in studies that combine CSS and CBR. Indeed, agent-based modeling is used extensively in public health [51]. Expanding the application and integration of these CSS techniques with CBR could significantly enrich the toolkit for tackling complex problems across various domains. For instance, agent-based models might offer a more granular understanding of individual behaviors within a system [52], while social network analysis could reveal the interconnectedness between various elements. Further exploration and integration of these methodologies into CBR could foster innovation, refinement, and enhance problem-solving capabilities.

The prevalence of business management as the field utilizing CSS and CBR topics is the most notable, as it signifies a trend in applying these methodologies to address complex organizational challenges. Businesses often grapple with intricate structures, interdependent variables, and dynamic systems. The adaptability of CSS techniques in modeling intricate relationships and predicting system behaviors that require CBR methods might explain the field's inclination towards these methodologies. However, the limited engagement from other domains prompts further inquiry—for example, why fields like environmental science or public health, which also deal with intricate systems, might not be adopting integrated CSS and CBR approaches at the same pace. Investigating the reasons behind this selective adoption could offer valuable insights into the tailoring of CSS techniques for broader interdisciplinary applicability.

The use of named entity recognition and dynamic topic modeling in our research aimed to promote understanding of the evolution and overlap of CBR and CSS over time. By applying named entity recognition, we were able to extract and identify key entities (e.g., country, discipline) and concepts related to CBR and CSS from a large corpus of academic research. This allowed us to capture the multifaceted nature of these fields and track their evolution over time. Dynamic topic modeling further enhanced our analysis by uncovering temporal patterns and trends in the topics and themes associated with CBR and CSS. Combined with named entity recognition, the use of dynamic topic modeling helped us to gain insights into the dynamic relationship between CBR and CSS and the factors that could be driving their convergence. While some researchers have examined this intersection in specific disciplines, such as Lawlor and colleagues' (2023) virtual Special Issue on systems science and community psychology, the use of these methods helped our study to extend beyond this scope by encompassing a broad spectrum of research areas using and applying CBR and CSS.

There were challenges in using named entity recognition and dynamic topic modeling to analyze the body of text. First, the corpus of text we analyzed encompassed diverse disciplines, leading to significant variations in language, terminology, and writing styles. This heterogeneity posed challenges for using standard named entity recognition models, which is why additional data cleaning and a novel named entity recognition model was used. The dynamic topic model presented challenges in understanding the temporal dynamics of topics related to CBR and CSS. This is because the evolving nature of these fields requires experimenting with the dynamic topic modeling formula to effectively track changes in topics over time. Finally, evaluating the performance and validity of both modeling approaches presented challenges, particularly in the absence of gold-standard datasets. Developing evaluation metrics (e.g., plots) and validation strategies (i.e., temporal resolution analysis) was crucial for assessing the reliability and accuracy of the extracted entities and dynamic topics.

*Limitations and Areas of Future Research*

While this scoping review offers comprehensive insights into the integration of CSS and CBR, several limitations must be acknowledged. One limitation lies in the inclusive but potentially broad definition of stakeholders, which might have influenced the representation and scope of the reviewed articles. The review's design and inclusion criteria

might have inadvertently overlooked or underrepresented certain domains or specific applications within CSS and CBR. Another limitation is the subjective nature of dynamic topic modeling's interpretation. While our topics have limited overlap, as determined by sensitivity analyses, the modeling output is still open to interpretation in how the words form the "topic". Next, while this study addresses the variability in prevalent topics over time, it does not explore external factors, such as historical, political, or regional influences, that might have shaped the evolution of CSS-CBR research. Further, while the initial literature search included multiple languages, the modeling techniques we used were limited to English manuscripts, introducing underrepresentation of global non-English research.

## 5. Conclusions

This scoping review identified the evolving dynamics between CSS and CBR, revealing their growing overlap and co-evolution. This study depicted the emergence of certain topics that drove the overlap between CSS and CBR, such as business management systems, food and land use and knowledge, water shed management, and social systems. The temporal analysis illustrated a clear shift in the subjects and themes over time, suggesting a progressive alignment between these two disciplines, notably around the late 1990s and early 2000s, with a decline and resurgence in their shared topics in subsequent years. The review identified the ascendance of system dynamics topics and modeling and highlighted the need for wider adoption and integration of other CSS methodologies like agent-based models and social network analysis into CBR. Together, these findings highlight opportunities to enhance future investigations and applications in studies that integrate CSS and CBR.

**Author Contributions:** Conceptualization, T.R.M. and R.W.S.C.; methodology, T.R.M., R.W.S.C. and N.C.; validation, T.R.M., R.W.S.C., N.C. and H.V.; formal analysis, T.R.M. and R.W.S.C.; investigation, T.R.M., R.W.S.C., N.C. and H.V.; data curation, H.V.; writing—original draft preparation, T.R.M.; writing—review and editing, T.R.M., R.W.S.C., N.C. and H.V.; visualization, T.R.M. and N.C.; supervision, T.R.M. and R.W.S.C.; project administration, T.R.M. and R.W.S.C.; funding acquisition, R.W.S.C. All authors have read and agreed to the published version of the manuscript.

**Funding:** The research reported in this publication was supported by the National Institute on Alcohol Abuse and Alcoholism of the National Institutes of Health under award number K01AA027564 to Coulter. Research reported in the publication was also supported by the National Institute on Child Health and Human Development under award number K99HD109456 to Moore. The funders had no role in the design of this study, its execution, analyses, interpretation of the data, or decision to submit the results. The content is solely the responsibility of the authors and does not necessarily represent the official views of the National Institutes of Health. Partial funding for open access was provided by Tufts University Hirsh Health Sciences Library's Open Access Fund.

**Data Availability Statement:** Publicly available manuscripts were analyzed in this study. These data can be found using manuscript databases and search engines such as PubMed and Google Scholar. The code utilized to run the models used in this manuscript can be found on Github.

**Acknowledgments:** We would like to acknowledge the invaluable contributions of Hayden Stec, Rohit Musuku, and Ramatu Abdul Hamid, students who helped to review manuscripts and discuss their inclusion in this study.

**Conflicts of Interest:** The authors declare no conflicts of interest.

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
