# Peer review of "The Integration of Complex Systems Science and Community-Based Research: A Scoping Review"

_systems, doi:10.3390/systems12030088_

Round 1

Reviewer 1 Report

Comments and Suggestions for Authors

Report on the manuscript: systems-2810544 

This work investigated the Integration of Complex Systems Science & Community- 2 Based Research: A Scoping Review.by Travis R. Moore , Nicholas Cardamone , Helena Vonville , and Robert W. S. Coulter. The following are some comments that the authors might like to take into account when revising the paper:

1.The English needs to be checked since there are problems with sentence structure and clause construction. Please kindly clarify the main background of this paper in the revision.

2.The author proposed Dynamic Topic Modeling in 2.8.2, but the following contents are not literal descriptions at all. Could the author provide physical characterization Modeling? , such as formulas, etc.

3In the full text, the author does not know what software to use to draw pictures, the pixels of the pictures in this paper are too poor, and the information in many pictures is blurred ,such as Fig.1. It is suggested that the author modify the saving mode, can you give a higher resolution picture?

4.    There are too many abbreviations in this article, which affects the reader to read the full text. It is recommended that the author abbreviate in the appropriate place, and do not abbreviate the whole text.

5. Some related works about stability analysis should be added to improve the quality of the literature review such as Output feedback stabilization of stochastic feed forward systems with unknown control coefficients and unknown output function,Stability analysis of semi-Markov switched stochastic systems. CS2Fusion: Contrastive learning for Self-Supervised infrared and visible image fusion by estimating feature compensation map. A note on global stability of a degenerate diffusion avian influenza model with seasonality and spatial Heterogeneity.

6.The author should give the source of the method in this paper in detail, and point out the problems solved by this method in this paper, and further compare with some recent papers, such as references [22,4] in this paper.

7.What is difficulty in analysis method of this article? Please give a remark to explain it. There are some typos the authors should check carefully in the coming version.

Comments on the Quality of English Language

 English very difficult to understand/incomprehensible

Author Response

Thank you for dedicating your time to reviewing our article and for offering thoughtful feedback. We appreciate the effort you have put into evaluating our work. Below, we will provide a detailed response to each of your comments and suggestions.

  1. The English needs to be checked since there are problems with sentence structure and clause construction. Please kindly clarify the main background of this paper in the revision.

We have carefully reviewed the English language and addressed any issues with sentence structure and clause construction. Additionally, we have provided further clarification on the main background of the paper in the revised version. Specifically, we have elaborated on the research context, objectives, and significance to enhance the reader's understanding of the study. We believe these revisions have strengthened the clarity and coherence of the manuscript.

  1. The author proposed Dynamic Topic Modeling in 2.8.2, but the following contents are not literal descriptions at all. Could the author provide physical characterization Modeling? , such as formulas, etc.

We have carefully considered your suggestion and have now incorporated a detailed description of the dynamic topic modeling formula, including the relevant mathematical formula, to provide a more concrete and literal characterization of the methodology.

  1. In the full text, the author does not know what software to use to draw pictures, the pixels of the pictures in this paper are too poor, and the information in many pictures is blurred ,such as Fig.1. It is suggested that the author modify the saving mode, can you give a higher resolution picture?

We have recreated Figures and uploaded them into the manuscript as .TIFF files so that they are higher resolution. We have also added short explanatory notes for Figures 2-4. We are also happy to provide all high-resolution figures and tables in a zip folder for the journal to insert in the paper on their end.

  1. There are too many abbreviations in this article, which affects the reader to read the full text. It is recommended that the author abbreviate in the appropriate place, and do not abbreviate the whole text.

We have carefully revised the text to ensure that abbreviations are used sparingly and only where appropriate. We have followed the recommendation to abbreviate selectively, ensuring that it does not overwhelm the readability of the full text. For example, we removed use of “NER”, and now have Named Entity Recognition written out. Additionally, we have provided explanations or definitions for any abbreviations used to facilitate comprehension for the reader. We believe these adjustments have improved the readability and accessibility of the manuscript while maintaining clarity and conciseness.

  1. Some related works about stability analysis should be added to improve the quality of the literature review such as Output feedback stabilization of stochastic feed forward systems with unknown control coefficients and unknown output function ,Stability analysis of semi-Markov switched stochastic systems. CS2Fusion: Contrastive learning for Self-Supervised infrared and visible image fusion by estimating feature compensation map. A note on global stability of a degenerate diffusion avian influenza model with seasonality and spatial Heterogeneity.

We agree that expanding the literature review to encompass relevant works on stability analysis would improve the quality of the manuscript. In response to your comment, we have included a section on sensitivity analysis in our manuscript, specifically focusing on temporal resolution analysis within the context of dynamic topic modeling. Temporal resolution analysis involves investigating the impact of different time intervals (e.g., yearly, quarterly, or monthly) on the stability and coherence of topics over time. By varying the granularity of the time scale, we assessed how the temporal resolution influences the evolution of topics. This analysis has been incorporated into Section 2.8.2 of the manuscript, where we discuss the methodology of dynamic topic modeling and the considerations for model parameterization. We cite the paper Kumar et al. (2011) as an example of authors who have performed this analysis.

  1. The author should give the source of the method in this paper in detail, and point out the problems solved by this method in this paper, and further compare with some recent papers, such as references [22,4] in this paper.

The sources of the dynamic topic model and named entity recognition have been cited in sections 2.8.1 and 2.8.2. Additionally, a new paragraph has been added at the end of section 4.0 to point out the problem solved by using this method in this paper and to compare it to the Lawlor et al 2023 paper. We do not compare our paper to our previously published paper because it is a protocol. The new paragraph is replicated below for ease of review:

“The use of Named Entity Recognition and Dynamic Topic Modeling in our research aimed to begin understanding the evolution and overlap of CBR and CSS over time. By ap-plying Named Entity Recognition, we were able to extract and identify key entities (e.g., country, discipline) and concepts related to CBR and CSS from a large corpus of academic literature. This allowed us to capture the multifaceted nature of these fields and track their evolution over time. Dynamic Topic Modeling further enhanced our analysis by uncover-ing temporal patterns and trends in the topics and themes associated with CBR and CSS. Combined with Named Entity Recognition, the use of Dynamic Topic Modeling helped us gain insights into the dynamic relationship between CBR and CSS and the factors that could be driving their convergence. While some researchers have examined this intersec-tion in specific disciplines, such as Lawlor and colleagues’ (2023) virtual special issue on systems science and community psychology, the use of these methods helped our study extend beyond this scope by encompassing a broad spectrum of research areas using and applying CBR and CSS.”

Reviewer 2 Report

Comments and Suggestions for Authors

Dear authors,

Thank you for the interesting topic of your submission focused on the interconnection of Complex Systems Science (CSS) and Community-Based Research (CBR).

In my opinion, the general focus on the linkage of two systemic approaches/concepts is the correct and original approach, especially considering its applicability across scientific fields.

From a methodological standpoint, I consider the article to be correctly implemented, including the conducted PRISMA review standard. The results of the studies are meaningfully presented, including discussion and outcomes.

The only minor flaw, which should be addressed in minor revisions, is the insufficient justification of the formulated research questions. Why does this issue have such research potential? What implications will the facts you have found have?

Comments on the Quality of English Language

The overall English quality is good, I have not found any substantial faults.

Author Response

Thank you for dedicating your time to reviewing our article and for offering thoughtful feedback. We appreciate the effort you have put into evaluating our work. Below, we will provide a detailed response to each of your comments and suggestions.

  1. The only minor flaw, which should be addressed in minor revisions, is the insufficient justification of the formulated research questions. Why does this issue have such research potential? What implications will the facts you have found have?

Thank you for this feedback. We have addressed this comment by adding a few clarifying sentences in the introduction. The new paragraph is reproduced below:

“While this example pertains to nutrition and public health, numerous instances can be found in diverse fields such as environmental science [17,18], healthcare [19], medicine [20], and food systems [21], among others. More recently, there are special journal issues being dedicated to the overlap between CSS and CBR [22]. Despite the growing utilization of these combined approaches, their deployment and impact remain relatively unex-plored, with no existing scoping review encompassing both CSS and CBR [4]. Exploring overlap between CSS and CBR can uncover synergies and complementarities between these two fields, leading to more integrated and holistic solutions to complex problems. Exploring the overlap between these fields can also inspire the development of novel re-search methodologies and tools that are tailored to address complex real-world challeng-es. A comprehensive search conducted on September 1st, 2019, across various databases yielded no scoping protocol or review specifically focusing on the integration of CSS and CBR, although a related systematic review explored community participation in health systems research [23]. However, this review did not treat CSS and CBR as distinct bodies of literature and primarily concentrated on health systems.”

Round 2

Reviewer 1 Report

Comments and Suggestions for Authors

Ok

Comments on the Quality of English Language

OK